# A ML-Based Resource Allocation Scheme for Energy Optimization in 5G NR

**DOI:** 10.3390/s25164978

**Published:** 2025-08-12

**Authors:** Xiao Yao, Antonio Pérez Yuste

**Affiliations:** ETSI Sistemas de Telecomunicación, Universidad Politécnica de Madrid, 28031 Madrid, Spain; xiao.y@upm.es

**Keywords:** 5G RAN, energy efficiency, resource allocation, RRC, machine learning

## Abstract

This paper proposes a machine learning (ML)-based energy optimization framework for 5G New Radio (5G NR) utilizing a Classification and Regression Tree (CART) algorithm. The methodology implements dynamic cell resource reconfiguration through predictive load forecasting, achieving a 42.3% reduction in energy consumption, while maintaining QoS parameters within 3GPP-specified thresholds. A case study with a network layout made up of an inter-band NR-NR Dual Connectivity (DC) was simulated to quantitatively validate our model.

## 1. Introduction

The next generation of 5G wireless networks is expected to provide massive capacity with ultra-high data rates, ultra-low latency, and extremely high reliability [1]. While consumer applications dominate public perception, 5G is equally transformative for industrial verticals, including healthcare, education, energy, mining, agriculture, and advanced manufacturing [2]. However, this technological leap comes at an energy cost. The exponential growth in 5G data traffic and the introduction of new use cases are driving an unsustainable increase in energy consumption, significantly raising the energy bill for Communication Service Providers (CSPs) [3]. And what is even more critical, this energy demand translates directly into higher carbon dioxide emissions [4] which cause, in turn, harmful effects to the environment. It is informed that about 2–4% of global carbon emissions are caused by Information and Communications Technologies (ICTs) [5].

Such a combination of rising operational costs and binding net-zero commitments has elevated energy efficiency to a strategic imperative for telecom operators [6]. According to the ‘Green Communications’ study in [4], the Radio Access Network (RAN) accounts for the largest share of power consumption in mobile communications, making intelligent resource allocation in 5G a critical challenge for achieving sustainable network operations.

In this paper, we present a power saving scheme based on an Artificial Intelligence (AI) decision tree ML model with a Classification and Regression Tree (CART) algorithm to shut off unnecessary radio cell resources when they are not required. Our AI is trained to dynamically determine the expected number of Data Radio Bearers (DRBs) required in a cell to satisfy the demand without degrading the user experience. Based on the prediction, the RAN orchestrator can decide in advance what radio resources should be activated or deactivated to optimize energy utilization.

To validate that scheme, we have developed a simple User Equipment (UE) stochastic traffic model by using a Radio Resource Control State Diagram (RRC-SD) to perform the dynamic behavior of one single user. Based on that, we have aggregated a number of UEs camped on a cell to estimate the probability density function (PDF) of required DRBs. This estimation identifies the necessary radio resources that must be kept and, consequently, provides a valuable input to decide what radio resources could be shut off. The estimated number of necessary DRBs so obtained is then compared to the real number of DRBs assigned by the RAN orchestrator for the AI to validate the model and reinforce its learning. Figure 1 represents the operational principle of our model.

The main contributions of this paper are summarized as follows:We propose a novel, multi-domain sleep-mode scheme for 5G RAN energy optimization;We design a statistically grounded UE traffic model based on a custom RRC-SD to simulate variant UE demand;We implement and evaluate a lightweight, interpretable ML model using a decision tree (CART) algorithm to predict optimal sleep states based on UE traffic.

This paper is organized as follows. In Section 2, the main strategies for energy optimization on 5G NR are reviewed, paying particular attention to shutdown techniques. In Section 3, the testbed of a generic inter-band NR-NR Dual Connectivity architecture is described. Such a testbed represents our case study and will serve to simulate the daily operation of a 5G network. In Section 4, a set of sleep states for every cell in the 5G testbed is introduced to later quantify the energy saving. In Section 5, we set up a dataset to firstly train the AI and later proceed with the validation. Due to the lack of real data, such a dataset was made up from scratch by simulating a broad variety of on-cell camping cases with data traffic generated through our RRC-SD stochastic model. In Section 6, the previous dataset is employed to check the accuracy of five different ML algorithms, resulting in the Decision Tree model with CART being the best option among them. This model is then analyzed in Section 7 to assess the energy-saving performance in our particular use case. Finally, the main conclusions are drawn in Section 8.

## 2. Background

Multiple energy-saving schemes have been developed for 5G NR to optimize power consumption without compromising network performance or connectivity requirements. A primary approach involves the dynamic deactivation of underutilized resources, which is commonly referred to as “sleep-mode” operation [7]. This technique can be implemented across multiple domains, including time, frequency, antenna, and power dimensions, to enhance energy efficiency [8].

In 5G NR, time-domain sleep mode, also known as symbol shutdown, enables gNBs to completely deactivate transmission during idle time slots when no user data traffic exists. This can be achieved by periodically reducing the transmission rate of common signals, such as synchronization signal blocks (SSBs), which facilitates deeper sleep states and significantly enhances network energy efficiency [9]. Consequently, this approach effectively eliminates wasteful power consumption from non-essential control signaling.

In the frequency domain, carriers and bandwidth parts (BWPs) can also be shut down dynamically according to real-time traffic [10]. Additionally, 3GPP has introduced the use of capacity booster cells to improve the energy efficiency of the RAN. When a candidate cell is turned off to conserve energy, the capacity booster cell is activated to provide alternative service and coverage, ensuring a seamless user experience [11].

For the antenna domain, Multiple-Input Multiple-Output (MIMO) techniques play a crucial role in 5G to optimize spectral efficiency and network capacity through spatial diversity and multiplexing. However, full MIMO capacity is not always required, so unnecessary subsets of antennas can be turned off to save energy [12]. Similarly, in the power domain, transmission power can be intelligently adjusted based on traffic conditions and channel quality [13] to further reduce energy consumption.

By simply shutting off radio resources, the power consumed by a base station can be reduced significantly [14]. However, this approach may lead to performance degradation. To mitigate that issue, ML/AI algorithms have been proposed to dynamically adjust network operations and resource allocations according to the real-time demand and the Quality of Service (QoS) required, reducing both the energy consumption and the carbon footprint [14,15].

Reinforcement learning (RL) has proven to be an effective approach for optimizing network energy consumption by learning control policies through interactions with dynamic environments [16], such as making decisions on the activation or deactivation of radio resources in response to varying traffic patterns, refined according to the some paper. For example, [17] combines Q-learning with advanced sleep mode (ASM), proposing a time-domain energy-saving technique to gradually put BS into different levels of sleep. The Q-learning model intelligently decides how long to remain in different sleep modes to obtain better energy-saving results without explicitly predicting traffic. This approach results in a 57% reduction in energy consumption. Similarly, for the frequency domain, [18] introduces an RL agent to predict user traffic behavior and proactively activates just enough carriers to meet data demand, hence minimizing unnecessary energy use while preserving QoS. Furthermore, supervised learning can also be very useful in energy saving. In [19], a Neural Antenna Muting is introduced to supervise a learning-based neural network able to predict the antenna configuration based on per-user channel and beamforming information, muting unnecessary antenna elements to minimize the energy consumption.

Building on the discussion of energy efficiency in 5G, this paper further explores a multi-domain radio resource shut-off scheme, integrating supervised learning models to dynamically adjust resource allocation for optimized energy savings.

## 3. 5G NR Testbed Model

### 3.1. Network Layout

In order to test our energy-saving scheme, we have set up a 5G NR use-case model, with an NR-DC layout, through a master gNB utilizing carrier aggregation (CA), along with a secondary gNB for Dual Connectivity (DC). The master gNB controls a primary cell using the n78 frequency band for both signaling and data traffic, while aggregating a secondary cell using the n28 frequency band to provide extensive coverage for data traffic only. The n78 band is divided into three sectors, each covering a 120-degree arc to provide complete coverage of the entire area. Meanwhile, the primary cell of the secondary gNB operates over a smaller area with higher capacity and lower latency, utilizing the mm-wave band n258. It is also worth mentioning that the selected layout and network features are specific to our testbed setup. It is representative of practical and commonly used 5G NR deployments, combining 5G low-band, mid-band, and high-band frequency bands. This configuration allows us to evaluate the proposed energy-saving model in a realistic yet controlled scenario, with the methodology being extensible to other deployment variations. The network layout is depicted in Figure 2.

The main features for each cell in our layout are shown in Table 1.

### 3.2. Network Resource Feature

As for the simulation of our model, we have taken two frames, which equal to 20 milliseconds, to represent the time–frequency resource grid allocation within the 5G network. We have configured 4800 resource elements (REs) per DRB. In this paper, DRB is used as a key figure to evaluate the network traffic in a cell. It represents a data pipe in the RAN that transports data packets over a defined route. In subsequent sections, the number of DRBs will be an essential indicator for the QoS that the network can provide and the UE can demand. Based on that, both the cell throughput and the UE throughput figures can be estimated.

We have divided the n78 cell into three sectors to balance the coverage and capacity as mentioned above. Additionally, we have divided the n258 cell into five different BWPs (n258-1, n258-2, n258-3, n258-4, and n258-5). For a more efficient allocation of spectrum, each BWP can be dynamically assigned to users based on real-time traffic demands, ensuring that high-priority services receive the necessary resources while maintaining robust connectivity for all users.

Furthermore, we use the multi-layer antenna system to enhance network capacity and performance. It is implemented through massive MIMO configurations, which enable better spatial multiplexing and improved signal quality. The detailed features for all resources in our network layout are summarized in Table 2.

## 4. Resource Allocation for Different Sleep Schemes

### 4.1. Resources Definition

Other contributions mainly focus on only one domain to deploy the sleep-mode scheme by shutting off unnecessary resources [9,10,11,12,13]. However, in our configuration, radio resources across time, frequency, and antenna domain are all considered together. We combine different resource shutdowns from different domains to configure the sleep-mode operation in each cell. The following radio resources were considered:*Slot*: Slot is a time-domain resource. The duration of a slot can vary based on the subcarrier spacing, which ranges from 15 kHz to 240 kHz. To describe the scheme of our sleep states clearly, we have considered 10 slots as the scheduling time.*Physical Channel*: Synchronization signal block (SSB) is essential for communication and must be enabled in all cells. The Physical Data Shared Channel (PDSCH), Demodulation Reference Signal, and Phase Tracking Reference Signal are all enabled in every cell to transfer data. However, the Physical Downlink Control Channel (PDCCH) is only enabled in cell n78 for signaling while being disabled in both n28 and n258 because these two cells are only used for data traffic.*Carrier*: Carrier resources belong to the frequency domain. We have defined three different carriers in our network model, as shown in Table 2.*Layer*: In the antenna domain, we configured that all cells use two antenna layers, except for the n78-2 cell, which utilizes four layers.

### 4.2. Resource Allocation in Each Cell

We have defined seven sleep states per layer for each cell, with different resources allocated for downlink and uplink data transmission. It is important to note again that the different resource allocation schemes for each sleep state in this section are arbitrarily designed to simulate our 5G deployment scenario.

The n28 frequency band uses FDD, allowing all slots to be used for downlink transmission. Sleep states of one layer in the n28 cell are shown in Table 3.

Differently, the n78 frequency band uses TDD, thus we use the first six slots for downlink data transmission while the other four slots are employed for uplink. We have allocated different numbers of slots in different states. Sleep states of the n78-x cell for one layer are shown in Table 4.

In the n258 cell, we allocate different BWPs and slots in different states, while putting the rest into sleep. The n258 sleep states for one layer are shown in Table 5.

A view of resource grids for two frames under different sleep state configurations is depicted in Figure 3. In the resource grids, the green parts labeled PDSCH are the resources allocated in the sleep state used for downlink data transmission. The dark blue parts without labeling are the deactivated resources.

Figure 3a shows the case of the resource grid at sleep state 3 in the n258 cell. It can be observed that only the first three BWPs carry data while the fourth and fifth BWPs are put into sleep. Also, only four slots are activated and carrying data traffic in PDSCH, while the rest are put into sleep without being used. There are SSBurst signals at the first slots of the grid while the PDCCH and Channel State Information–Reference Signal (CSI-RS) are disabled. Figure 3b shows the resource grid of sleep state 4 in n78-2 cells. The n78-2 cell uses the second BWP of the whole n78 frequency band (middle BWP in the grid). Only four slots in the second BWP are activated and carrying data traffic in PDSCH, while the rest are put into sleep without being used. Since the n78 cells are used for signaling, the PDCCH and CSI-RS are enabled.

### 4.3. Sleep State Performance

According to the resource allocation scheme defined before, we compute the performance in every sleep state for each cell. The performance for each sleep state presented in this section is based on computational analysis using the selected layout and resource configurations, rather than derived from field measurements. Once again, we emphasize that the sleep states were arbitrarily defined to simulate a realistic deployment scenario. The goal is to estimate the potential energy savings associated with each sleep state under the assumptions made in our 5G NR model. Here, we just show the n258 cell as an example.

The power in Table 6 represents the relative power compared to the full-on downlink transmission power consumption in one layer. According to the network model, the n258 cell has two antenna layers. Thus, in layer 1 state 4, which is labeled as L1S4, the maximum number of DRBs that can be used in the simulated time interval is 635.

We consider that every layer has the same sleep states and configurations. Thus, the relative downlink power for layer x and state y (labeled as LxSy) can be computed as(1)$$\frac{\left(x-1\right)+relative\;power\;in\;y}{numbers\;of\;antenna\;layers\;of\;the\;cell}$$

## 5. UE Demand

To train and validate the AI-based energy optimization model, a dataset reflecting the UE traffic demand is required. However, in the absence of real-world traffic data from a live 5G network, we generated a synthetic dataset to model UE traffic demand. For this purpose, we developed a UE traffic model referred to as RRC-SD, allowing us to simulate realistic UE behavior under varying network load conditions. The resulting dataset quantifies UE demand in terms of the number of DRBs, as discussed in Section 3.

The DRB usage figures presented in this section reflect hypothetical—but realistic—traffic patterns and are intended to be treated as real by the AI model. This synthetic approach allows us to evaluate the potential of the proposed energy-saving scheme in a controlled yet practically meaningful environment.

### 5.1. UE Model Using RRC Diagram

To provide a dataset for training and validation for our AI model, a UE traffic model was developed, as shown in Figure 4.

When any user enters the 5G network in our model, i.e., it is switched on, it is firstly moved to the RRC-Idle state, where no dedicated DRBs are established. We consider that every UE can have a transition action in a time interval, δ. Once connected, the UE can be in various RRC-Connected states depending on the number of established DRBs and signaling radio bearers (SRBs). In these states, every UE can progress from one to a number “n” of DRBs depending on the REs required. Furthermore, UE can also transit back to the RRC-Idle state, meaning RRC release, or can also transit to the RRC-Inactive state, meaning RRC suspension. In the RRC-Inactive state, the UE is in a low-power mode with minimal signaling.

Once a UE enters the network, it initially resides in the RRC_ Idle state. After a random period of time, it changes to being connected and begins moving between all RRC states with different transition probabilities. When there are a number of camped users on a cell high enough, according to the theory of Markov chains, the total demand of DRBs in the cell will tend to be stable over time.

According to the general RRC-SD scheme depicted in Figure 4, we define the next six different RRC states:RRC state 1: RRC-Idle;RRC state 2: RRC-Inactive;RRC state 3: RRC-Connected with one DRB;RRC state 4: RRC-Connected with one DRBs;RRC state 5: RRC-Connected with three DRBs;RRC state 6: RRC-Connected with four DRBs.

Thus, the RRC state diagram can be mathematically modeled as a transition matrix for every user, which shows the probabilities of transitioning from one state to another. Each row represents a current state, and each column represents a possible next state.(2)$$\left[\;\begin{array}{cccccc} 1-\alpha & 0 & \alpha & 0 & 0 & 0\\ \gamma & 1-\gamma -\upsilon & \upsilon & 0 & 0 & 0\\ \beta & \rho & 1-\rho -\beta -\lambda_{1} & \lambda_{1} & 0 & 0\\ 0 & 0 & \;\mu_{1}\; & 1-\mu_{1}-\lambda_{2} & \lambda_{2} & 0\\ 0 & 0 & 0 & \mu_{2} & 1-\mu_{2}-\lambda_{3} & \lambda_{3}\\ 0 & 0 & 0 & 0 & \mu_{3} & 1-\mu_{3} \end{array}\;\right]$$

To provide more realistic user traffic data, it is necessary to simulate UE with different activity patterns. As for convenience, we have created three types of users with different transition probabilities: 30% of type A users, which represent a type of hungry-data-consuming user with more frequent transitions between high data transmission states (RRC-Connected with three or four DRBs); 50% of type B users, which represent a type of normal-data-consuming user with a moderate data transmission and occasional transitions to high data states; and 20% of type C users, which represent light-data-consuming users, mostly in RRC-Idle or RRC-Inactive states, with occasional low data transmission. Table 7 summarizes different transition probabilities for each user type.

Building on the previous general description, we specify the configurations for each cell in our network model layout. In n78-1 and n78-3 cells, we arbitrarily define three different states for RRC_CONNECTED. That is, we set the last connected state to void in the general RRC-SD of Figure 3. To achieve that, we set λ_3_ and μ_3_ to 0 in Table 7. In the n78-2 cell, we arbitrarily define two different states for RRC_CONNECTED. That is, we set the last two connected states to void in the general RRC state diagram, while we set *λ*_2_, *λ*_3_ and *μ*_2_, *μ*_3_ to 0. And, finally, in the n258 cell, we define four different states for RRC_CONNECTED, just the same as the general RRC state diagram.

The DRB usage in different RRC-Connected states also varies from different cell areas, as shown in Table 8. Network operators can easily modify these configurations to offer a varied QoS to UEs in their corresponding service area.

### 5.2. UE Distribution

To proceed with the simulation, we configured the distribution of UE populations across various cell areas based on the geographical layout of our network model. We increased the user count by steps of 50. In the n78-1 and n78-3 cell areas, the number of users ranges from 50 to 400, while in the n78-2 cell area, the UE population ranges from 50 to 600. The n258 cell offers high capacity, so we can consider it as a densely populated area, with the number of users varying between 50 and 1,400, hence, resulting in a total of 21,504 unique UE distribution combinations, enabling the model to generalize over a wide variety of spatial traffic scenarios.

### 5.3. UE Demand Results

We set the total simulation time to 500 time-units, with one time-unit equal to one frame (10 ms). Every simulation was run for every distribution scenario, and the UE demand traffic data was stored for each one. All our results show that the DRB usage, in each cell, tends to be stable over time in all scenarios, verifying that our RRC-SD model represents a Markov chain. In addition, we checked that the DRB usage in each cell approaches a normal statistical distribution quite well.

Figure 5 below shows the particular result obtained for the most crowded scenario, where the UE distribution is [400, 600, 400, 1400] users in each cell. That is, 400 in n78-1, 600 in n78-2, 400 in n78-3, and 1400 in n258. We refer to this as the “worst scenario”. As shown in Figure 5, the DRB usage tends to be stable after a short period. Thus, we selected the [50, 500] time interval to present the statistical results.

Since the volume of information is too huge, we have just chosen the traffic forecasted in the n258 cell for the worst (most crowded) scenario as an example. The probability density function (PDF) and cumulative distribution function (CDF) for this case are shown in Figure 6a and Figure 6b, respectively. We can observe how the DRB usage distribution approaches a normal distribution. We have also marked the 99th-percentile value of the distribution, which represents the number of DRBs required to satisfy the traffic demand of at least 99% of users.

### 5.4. UE Demand Detaset

Following the simulation, we recorded the mean and standard deviation of the next figures: DRB usage, cell throughput, and UE throughput for each cell across all distribution scenarios within our demand dataset. These data can be used in future studies for the reproduction of the required UE demand distribution. Additionally, we stored the 99th-percentile value of these metrics, which indicates that 99 percent of user demands can be satisfied, for subsequent decision-making purposes.

In our dataset, the 99th-percentile value of the DRB usage is an important indicator. With this data, we can determine the number of DRBs that the RAN orchestrator should allocate, as shown in Figure 1, and then determine the sleep state that our RAN testbed should be in. We stochastically select 70% of the dataset for training the AI model, which is designed to predict the appropriate sleep states as output for the 5G RAN under varying user distribution scenarios. The remaining 30% of the data is reserved for validation, where the model’s predicted sleep states are compared against the reference sleep states determined by the RAN orchestrator.

## 6. ML Model and Evaluation

### 6.1. Data Process

Starting from the 99th-percentile value of the UE DRB demand in each cell as the main feature in the entire dataset, along with the corresponding user number vector, a post-processed dataset in the format of [UserNumCase1, UserNumCase2, UserNumCase3, UserNumCase4, 99-percentile of DRB demand in n28, 99-percentile of DRB demand in n78-1, 99-percentile of DRB demand in n78-2, 99-percentile of DRB demand in n78-3, 99-percentile of DRB demand in n258] was set up as a baseline.

Then we matched the DRB usage data to their corresponding sleep states using the matching method explained in Section 4 to obtain the layer and sleep mode for the n28, n78-1, n78-2, n78-3, and n258 cells. As a result, we obtained the dataset for the machine learning model with the following format: [UserNumCase1, UserNumCase2, UserNumCase3, UserNumCase4, LxSy in n28, LxSy in n78-1, LxSy in n78-2, LxSy in n78-3, LxSy in n258].

The subset of the vector [UserNumCase1, UserNumCase2, UserNumCase3, UserNumCase4] represents the UE population in the n258cell, n78-1 cell, n78-2 cell, and n78-3 cell, respectively. This vector is later used as the input feature of our ML model. For a more concise expression, we refer to them as a vector of features [F1, F2, F3, F4].

According to the geographical layout of our 5G RAN model, the usage of DRBs in each cell is affected by the UE population in different cell areas. The DRB usage in the n28 cell is affected by the number of users in cells n78-1, n78-2, and n78-3, which are F2, F3, and F4 features. The DRB usage in the n78-2 cell is affected by the features F3 and F1. The DRB usages in n78-1, n78-3, and n258 cells are affected by their own cell area, which corresponds to features F2, F4, and F1, respectively. Therefore, we set an ML model for each cell with its corresponding input features to predict its DRB usage.

Finally, we used the holdout validation method, reserving 30% of the data in the dataset for validation, while using the remaining 70% of the data to train the model.

### 6.2. ML Model Selection for Our Case

Since our data was labeled based on the expected sleep states with different UE populations, we tested commonly used supervised learning models. Given the multi-class nature of our classification task, it was crucial to ensure that the selected ML models were well-suited for this specific scenario.

Another key consideration was computational efficiency. With a sampling interval of 500 time-units (equivalent to one unit per frame or 5 s in total), we needed to ensure that the chosen model could operate efficiently within this time constraint. Therefore, we evaluated five different classic ML models, as shown in Table 9, and we analyzed their performance independently.

Considering both accuracy and computation time obtained in training and validation, the ML model with the best performance indicators among all five was the decision tree (DT) model with the Classification and Regression Tree (CART) algorithm. Consequently, this model was the selected candidate for the AI to be integrated into the original 5G RAN scheme, in Figure 1, for energy optimization.

### 6.3. Model Analysis and Evaluation

Our selected model is a simple Classification Tree based on the CART algorithm. Considering our application scenario—which is relatively structured and well-defined—and the simplicity of the input features, we intentionally chose a lightweight and interpretable model.

To better understand the model’s learning behavior, we generated learning and validation curves by training the model on increasing subsets of data, ranging from 10% to 100%. For each subset size, a new random sample of the dataset was extracted to ensure diversity in training data. This sample was then consistently divided using a 70/30 train–validation split. To maintain model independence, each of the five cells were trained separately using their own randomly sampled subsets, meaning that the data used to train one cell model was completely independent from the subsets used for the others. This approach simulates cell-specific traffic patterns and avoids data leakage between models, ensuring that the performance evaluation reflects each cell’s ability to generalize based on its own traffic characteristics. The average training and validation accuracy across all five cells is shown in Figure 7. The results demonstrate stable performance and minimal overfitting, with training and validation accuracy remaining consistently close, which indicates a good generalization capability.

We also explored key hyperparameters of the decision tree, including the MinLeafSize and MaxNumSplits. While MinLeafSize showed minimal impact across a reasonable range (5–50), MaxNumSplits influenced accuracy in cases where it was set too low, leading to underfitting (the results can be found in Appendix A), so we kept the MaxNumSplits as the default, which does not limit the number of splits and keeps the MinLeafSize to 5. These findings support the conclusion that our selected model is both efficient and robust under the given problem structure. In order to better evaluate our selection, we present some analyses in the following subsections.

#### 6.3.1. Interpretability and Sensitive Analysis

Our classification trees are quite easy to interpret. Figure 8 is an example of the visual CART decision tree trained for the n78-1 cell. The tree clearly illustrates the step-by-step logic that the model uses to assign sleep states based on input features. In this case, only a single feature (F2) is used throughout the entire tree, indicating that F2 fully determines the classification in this configuration. This leads to a highly transparent model, where decisions can be directly traced to threshold comparisons on a single variable. It is worth noting that, in this study, classification models were developed using the MATLAB R2023b fitctree function, and decision tree structures were visualized using the MATLAB built-in function, which does not support programmatic customization of the class labels in the graphical mode. Therefore, class outputs are shown using numeric identifiers rather than descriptive labels. More details about classifications can be found in the confusion matrixes for each cell in the Appendix A.

Across all five cells analyzed, the dominant features influencing classification vary depending on the traffic layer. Table 10 summarizes the normalized feature importance for each cell. These results reinforce the flexibility and explainability of CART models for energy-saving behavior modeling in 5G RAN.

#### 6.3.2. Complexity and Scalability Analysis

The proposed CART-based ML scheme offers high computational efficiency and strong scalability in both inference time and data adaptability. As shown in Table 11, the number of nodes in each trained decision tree varies across cells, but all models remain relatively shallow. In practice, energy-saving decisions in operational 5G networks are typically made at intervals of at least 30–60 s, as excessively frequent transitions may lead to increased signaling overhead, instability, and diminished overall energy gains. As shown in Table 9, the total computational time for our model is 0.22 s, confirming its real-time feasibility.

Furthermore, the scalability of our approach is reinforced by the design of our UE demand dataset, which simulates a wide range of user distribution scenarios across the four main cell areas in our testbed. This extensive dataset enables the model to generalize effectively across diverse spatial traffic conditions.

These characteristics make the proposed framework not only computationally efficient but also highly suitable for real-world integration into practical 5G RAN architectures.

#### 6.3.3. EE- and QoS-Sensitive Accuracy

Due to the large number of classes in our problem, traditional metrics such as the F1-score, precision, or recall do not fully capture the impact of near-miss misclassifications. For instance, misclassifying from state 5 to 6 is very different from a mistake between state 5 and state 1 in terms of operational cost—yet both are treated equally in standard metrics. To address this, we draw inspiration from binary classification metrics and propose an adapted formulation of QoS-sensitive and Energy Efficient (EE)-sensitive accuracy, which considers the direction of misclassification.

In our resource allocation scheme, each sleep state offers a different level of DRBs to provide service to meet the UE demand. But except for the n258 cell, the difference in DRBs provided in neighboring states in each cell is small. Thus, if an instance is classified into one state higher than it should be, the network can still provide enough service to meet the UE demand but waste a little more energy. On the contrary, if an instance is classified into one state lower than it should be, the network can save more energy but might not meet 99% of the UE demand. Based on this, we propose a QoS-sensitive accuracy and an EE-sensitive accuracy to evaluate our model.

To better explain our proposed accuracy metrics, here we take the confusion matrix of the n28 cell as an example.

First, we consider each state combined with its higher state to form a binary classification case as shown in Figure 9. We can simplify the binary case as Figure 10 below. In this context, class *i* represents a specific sleep state or resource allocation state of the cell, where a certain number of DRBs are provided to meet user demand. class *i* + 1 represents a higher state that offers more resources. We focus on class *i* to evaluate how often traffic demand cases are correctly classified into this state or misclassified into its immediate higher or lower neighboring states, which directly affects either energy consumption or service quality.

The TPi means the number of instances that are correctly classified into class *i*. The FNi is the number of instances that should be classified into class *i* but are wrongly classified into one state higher in class *i* + 1. FPi is the number of instances that should be classified into class *i* + 1 but are wrongly classified into one state lower in class *i*. The TN_i_ is the TP_i+1_ in the next state. With this, we can form the formula of our proposed accuracies.

For the QoS-sensitive accuracy, if an instance was classified into a higher class, it is acceptable since it provides more DRBs to give a more qualified service. We can obtain this value as below:(3)$$QoS\;Acc=\;\frac{\sum {TP}_{i}+\sum {FN}_{i}}{total\;samples}$$

Similarly, the EE-sensitive accuracy can be present as below:(4)$$QoS\;Acc=\;\frac{\sum {TP}_{i}+\sum {FP}_{i}}{total\;samples}$$

Table 12 shows the EE- and QoS-sensitive accuracy for each cell. More over, these two accuracy figures can be chosen by the operator depending on their actual need, whether to save more energy with a little damage to user experience or waste a little more energy to meet all the UE demands. Using these accuracies, the operator can further optimize the whole model.

#### 6.3.4. Comparison with the Existing Methods

Although we did not implement reinforcement learning (RL)-based resource allocation schemes in this study, we emphasize that our proposed method addresses a different dimension of the energy-saving problem. Most existing studies focus on policy optimization for binary cell on/off switching or single-domain resource control. In contrast, our approach introduces a multi-domain, fine-grained sleep-mode scheme, allowing partial deactivation of radio resources in time, frequency, and antenna domains, which is rarely addressed in existing energy-saving schemes.

Our proposed scheme has been developed and evaluated under a custom-designed 5G RAN testbed, which models realistic network conditions, diverse UE distributions, and DRB demand patterns using a configurable RRC-based behavior model. Due to the unique modeling assumptions and the fine-grained, multi-domain sleep-mode design, our framework does not map directly onto existing simulation environments or resource allocation schemes, especially those based on traditional on/off cell control or reinforcement learning. As a result, a direct numerical comparison with prior studies is not straightforward.

#### 6.3.5. Open Dataset

Our synthetic dataset, named 5G-RRC-SD-UE, has been published openly via Zenodo and is freely available to the research community. This dataset is intended to support ML-based energy optimization research in 5G RAN and can be accessed openly. By making our dataset publicly available, we aim to promote transparency, reproducibility, and encourage further research in this area.

As we discussed in the previous section, due to the customized 5G NR testbed and the multi-domain sleep-mode scheme applied in this study, direct comparison with existing approaches is not straightforward. Nevertheless, 5G-RRC-SD-UE provides researchers with the opportunity to develop alternative strategies and evaluate them against our results using the same UE demand scenarios. We believe this dataset will serve as a useful benchmark for future studies in energy-efficient 5G network management.

## 7. Results

### 7.1. ML Model Accuracy Results

Table 13 shows the training and validation accuracy in each cell. In our simulation, the n258 cell has the best performance with a validation accuracy of 97.7% while the rest of cells have results around 80% of validation accuracy. Taking the average of the 5 cells’ accuracy results, we obtained an overall training accuracy for our decision tree model of 85.1%, and an overall validation accuracy of 84.7%.

### 7.2. Energy-Saving Results

Due to the lack of real UE population data to test our ML-based energy saving scheme, and considering the large number of UE distribution scenarios in our dataset, we have selected only two user scenarios from our validation dataset to illustrate the energy saving achieved. These are the UE distributions: [250, 300, 200, 800] and [400, 600, 400, 1400]. They represent, respectively, an average case, named as “normal scenario”, and a crowded case, named as “worst scenario”. Using the formula shown in Section 4 to compute relative power consumption, the power saving results for both scenarios were computed. We took power figures for every cell as defined in Section 3 before.

The final results achieved with our Decision Tree CART algorithm resource allocation scheme for energy saving were quite promising. The normal scenario consumes 57.7% of the full-on downlink transmission power consumption, thus saving 42.3% of power, while the worst scenario saves 6.69% of the full-on downlink transmission power consumption.

### 7.3. Limitations and Future Work

This study relies exclusively on synthetic traffic data generated using our proposed RRC-SD stochastic user behavior model. While the UE demand dataset is theoretically grounded with a novel RRC-based traffic model and incorporates a diverse mix of UE profiles with distinct transition probabilities, this study is limited due to the fact that synthetic datasets cannot fully capture the complexity, burstiness, and variance present in real-world operational 5G networks. Another important limitation of our current study is that our synthetic dataset simulates UE demand under the user distributions in our specific 5G-RAN layout; it does not account for diurnal patterns commonly observed in real networks—such as peak hours, nighttime inactivity, or special event-driven load surges.

Regarding these limitations, our dataset is designed to reflect a wide variety of user distribution scenarios, with UE population combinations spanning sparse to highly crowded conditions. This range allows the model to generalize across diverse spatial load patterns, making it highly applicable to real-world 5G networks. For future work, we are exploring partnerships with network operators to access real RAN logs or KPI data and to deploy and test our sleep-mode scheme in controlled environments to validate our UE demand model and our ML-based multi-domain sleep strategy.

## 8. Conclusions

This paper proposes an energy consumption optimization scheme based on a decision tree ML model with the CART algorithm. A generic 5G inter-band NR-DC architecture was introduced as a testbed to validate the approach. On top of that, we defined a sleep-mode scheme, considering multi-domain radio resource shut off for the 5G NR. Then, we configured a UE profile and generated a UE demand dataset in every cell through a newly proposed RRC-SD stochastic model. Using the dataset, we trained five ML models, obtaining a validation accuracy of 84.7% to dynamically predict the corresponding RAN sleep states based on different UE demands. Our ML-based multi-domain resource allocation scheme achieved 42.3% energy consumption reduction in the normal UE demand scenario.

Compared to previous studies that primarily focus on ML-based single-domain radio resource management, our work considers multi-domain radio resource coordination, offering a more comprehensive approach to energy saving. In addition, we address the limitation of lacking real-environment data by proposing a novel RRC-SD model to generate a synthetic UE traffic dataset. While this dataset effectively enables training and evaluation of our scheme, it is not as varied as the UE traffic observed in real-world scenarios. As future work, we plan to incorporate real-environment UE traffic data to further validate our proposed sleep scheme in real 5G RAN deployments. Furthermore, we can explore the use of reinforcement learning models for our proposed energy-saving scheme in more diverse and dynamic real-world environments.

## Figures and Tables

**Figure 1 sensors-25-04978-f001:**
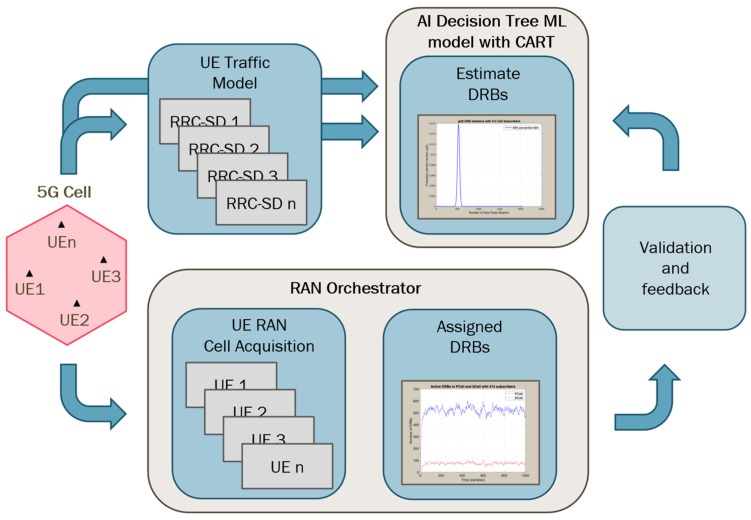
Operational principle of the proposed model.

**Figure 2 sensors-25-04978-f002:**
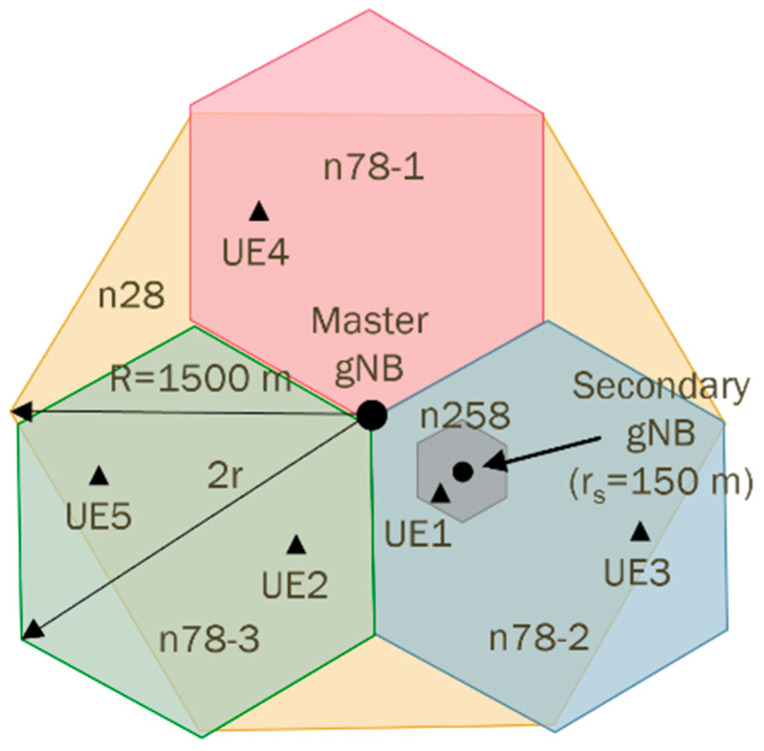
**5G** NR network model layout.

**Figure 3 sensors-25-04978-f003:**
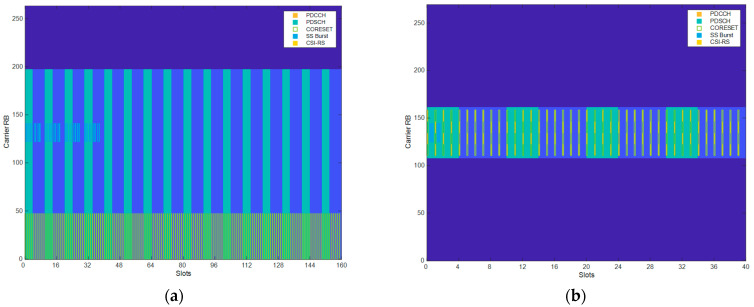
Examples of resource grid using our proposed sleep schemes: (**a**) sleep state 3 in n258 cell; (**b**) sleep state in n78-2 cell.

**Figure 4 sensors-25-04978-f004:**
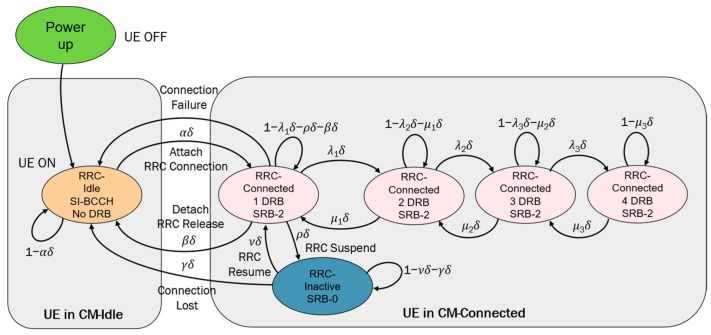
The general RRC state diagram (RRC-SD) for our UE model.

**Figure 5 sensors-25-04978-f005:**
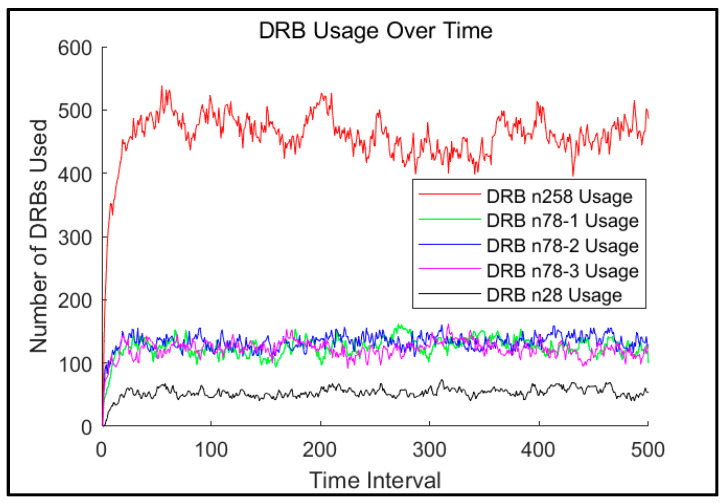
DRB usage of the n258 cell along time.

**Figure 6 sensors-25-04978-f006:**
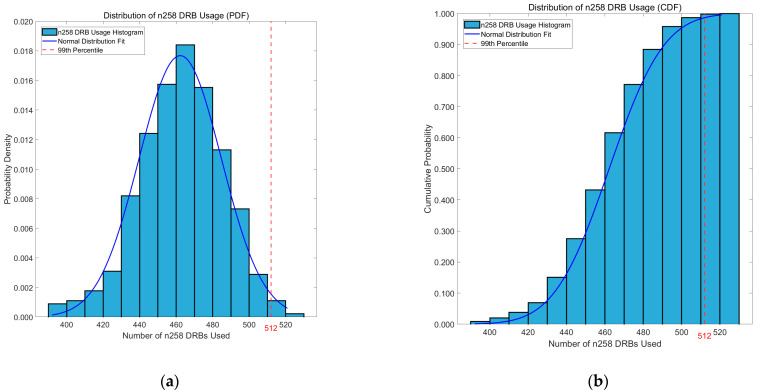
DRB usage distribution of the n258 cell in the worst scenario: (**a**) PDF distribution; (**b**) CDF distribution.

**Figure 7 sensors-25-04978-f007:**
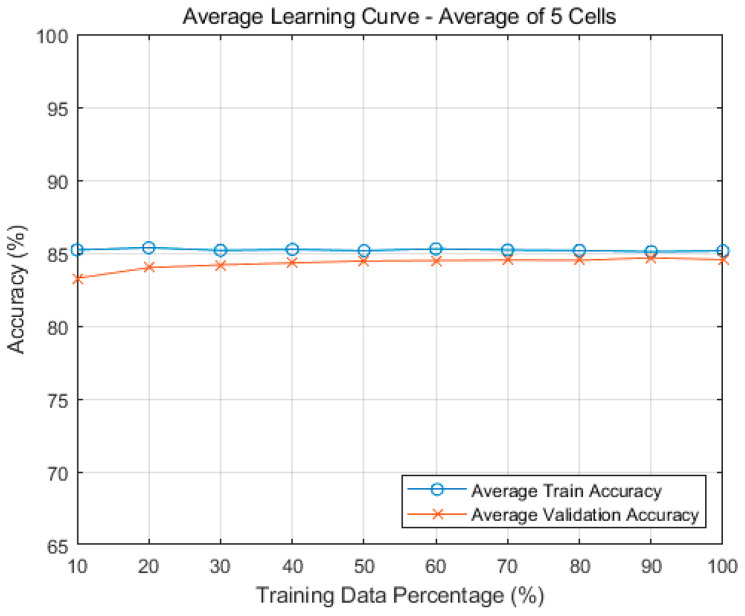
Average learning and validation curve for our ML model.

**Figure 8 sensors-25-04978-f008:**
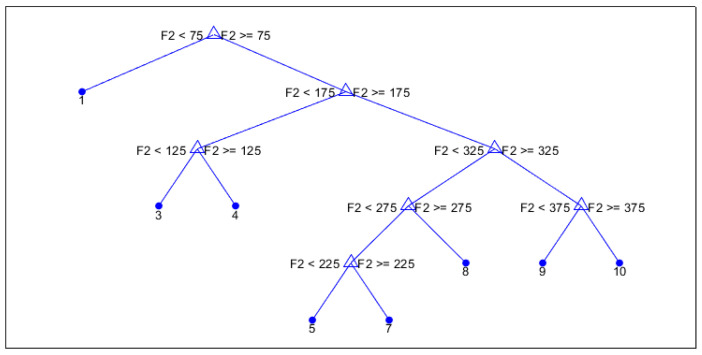
Visual decision tree trained for the n78-1 cell.

**Figure 9 sensors-25-04978-f009:**
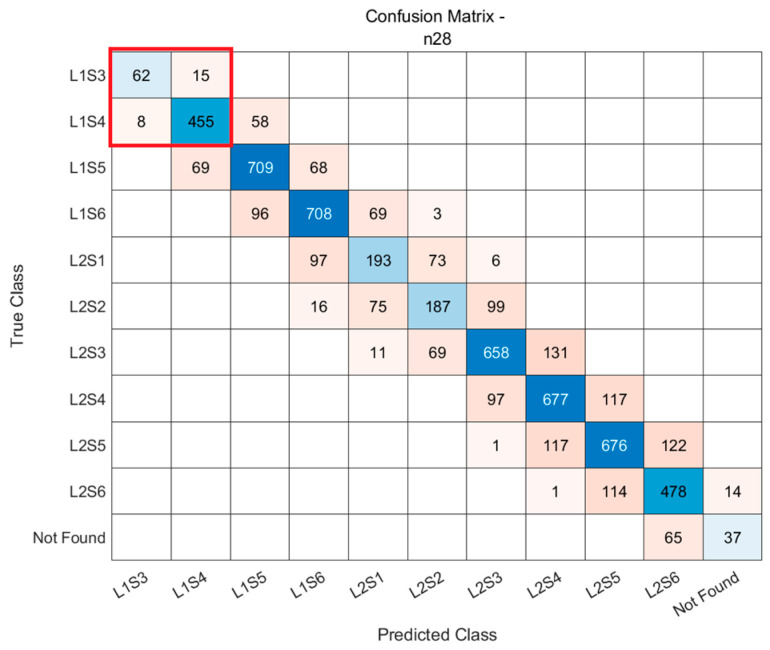
Confusion matrix of n28 cell.

**Figure 10 sensors-25-04978-f010:**
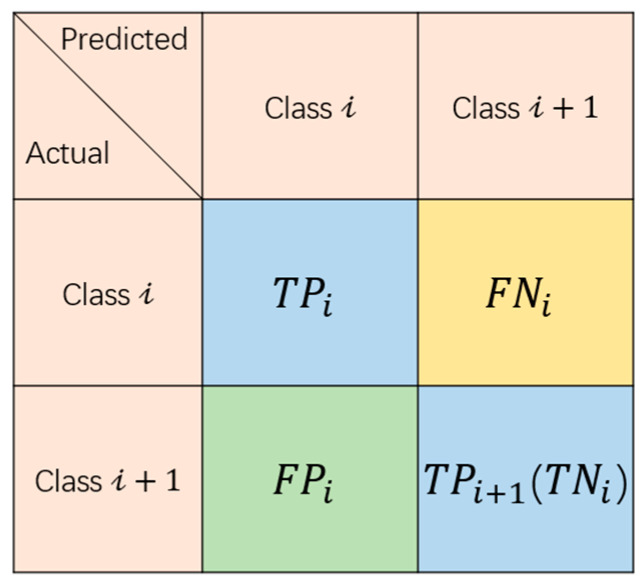
The binary confusion matrix for class *i*.

**Table 1 sensors-25-04978-t001:** Features of every cell in the 5G RAN testbed.

Frequency Band	BW (MHz)	TDD/FDD	SCS (kHz)	µ	Slots/ Frame	RBs	Antenna
n28	10	FDD	15	0	10	52	omni
n78	100	TDD	30	1	20	273	sector
n258	400	TDD	120	3	80	264	omni

**Table 2 sensors-25-04978-t002:** Features of resources in the 5G NR testbed.

Resources	BW (MHz)	BWP	RBs	Slots/ Frame	Layers	Antenna	Power (W) ^1^
**n28**	10	1	52	10	2	Omni	5
**n78-1**	40	1	109	20	2	Sector	5
**n78-2**	20	2	55	20	4	Sector	5
**n78-3**	40	3	109	20	2	Sector	5
**n258-1**	120	1	79	80	2	Omni	3
**n258-2**	100	2	66	80	2	Omni	
**n258-3**	80	3	53	80	2	Omni	
**n258-4**	60	4	40	80	2	Omni	
**n258-5**	40	5	26	80	2	Omni	

^1^ the power values listed are derived from empirical configurations and practical deployment experience in our testbed. They are intended to reflect realistic operational parameters but not standardized.

**Table 3 sensors-25-04978-t003:** Sleep scheme for n28 cell.

State	Slot									
	0	1	2	3	4	5	6	7	8	9
**0**										
**1**	DL									
**2**	DL	DL								
**3**	DL	DL	DL	DL						
**4**	DL	DL	DL	DL	DL	DL				
**5**	DL	DL	DL	DL	DL	DL	DL	DL		
**6**	DL	DL	DL	DL	DL	DL	DL	DL	DL	DL

**Table 4 sensors-25-04978-t004:** Sleep scheme for n78-x cell.

State	Slot									
	0	1	2	3	4	5	6	7	8	9
**0**										
**1**	DL						UL			
**2**	DL	DL					UL			
**3**	DL	DL	DL				UL	UL		
**4**	DL	DL	DL	DL			UL	UL		
**5**	DL	DL	DL	DL	DL		UL	UL	UL	
**6**	DL	DL	DL	DL	DL	DL	UL	UL	UL	UL

**Table 5 sensors-25-04978-t005:** Sleep scheme for n258 cell.

State	BWP					Slot				
	1	2	3	4	5	0	1	2	3	4
**0**										
**1**	DL					DL	DL			
**2**	DL	DL				DL	DL	DL		
**3**	DL	DL	DL			DL	DL	DL	DL	
**4**	DL	DL	DL	DL		DL	DL	DL	DL	DL
**5**	DL	DL	DL	DL	DL	DL	DL	DL	DL	DL
**6**	DL	DL	DL	DL	DL	DL	DL	DL	DL	DL

**Table 6 sensors-25-04978-t006:** Energy-saving performance for n258 cell.

State	BW (MHz)	DL Slots	UL Slots	Relat. Power	PDSCH (kREs/s)	PDCCH (kREs/s) ^2^	DRB (bearer/s)	DRB in Grid
**0**	28.90	0	0	0.02	0	0	0	0
**1**	113.88	2	2	0.12	20,476.8	0	4266	85
**2**	208.92	3	3	0.28	55,248.0	0	11,510	230
**3**	285.48	4	4	0.50	101,155.2	0	21,075	422
**4**	343.20	5	5	0.75	152,352.0	0	31,740	635
**5**	380.76	5	5	0.83	169,200.0	0	35,250	705
**6**	380.76	6	6	1	203,054.4	0	42,303	846

^2^ PDCCH throughput in n258 cell is always 0, because in our network model definition, this cell is just used for data traffic, so we disabled the PDCCH signal. Similarly, we defined the n28 cell.

**Table 7 sensors-25-04978-t007:** Transition probabilities of UE.

Transition	RRC Action	Type A (30%)	Type B (50%)	Type C (20%)
**α**	Attach	0.14	0.08	0.06
**β**	Detach	0.18	0.18	0.18
**λ_1_**	DRB+	0.10	0.06	0.03
**λ_2_**	DRB+	0.04	0.03	0.02
**λ_3_**	DRB+	0.02	0.02	0.01
**μ_1_**	DRB−	0.14	0.16	0.18
**μ_2_**	DRB−	0.16	0.18	0.20
**μ_3_**	DRB−	0.18	0.20	0.22
**ρ**	Suspend	0.30	0.30	0.30
**γ**	Release	0.04	0.04	0.04
**υ**	Resume	0.14	0.08	0.06

**Table 8 sensors-25-04978-t008:** DRB usage in each cell area.

Cell	Connected State 1	Connected State 2	Connected State 3	Connected State 4
**n78-1**	1 n78-1 DRB	2 n78-1 DRB	2 n78-1 DRB 1 n28 DRB	Void
**n78-2**	1 n78-2 DRB	1 n78-2 DRB 1 n28 DRB	Void	Void
**n78-3**	1 n78-3 DRB	2 n78-3 DRB	2 n78-3 DRB 1 n28 DRB	Void
**n258**	1 n258 DRB	2 n258 DRB	3 n258 DRB	3 n258 DRB 1 n28 DRB

**Table 9 sensors-25-04978-t009:** Comparison of ML models.

ML Model	Training Accuracy	Validation Accuracy	Computation Time (s)
**Decision tree (CART)**	85.06%	84.73%	0.22
**Decision tree (ID3)**	85.06%	84.73%	1.20
**KNN (k = 4)** ^3^	84.17%	84.01%	0.53
**Random forest**	85.04%	84.72%	6.60
**RBF SVM**	85.06%	84.74%	13.84

^3^ the reason to choose the k factor as 4 can be seen in the Appendix A.

**Table 10 sensors-25-04978-t010:** Normalized feature importance of 5 cells.

Cell	F1	F2	F3	F4
**n258**	1	0	0	0
**n78-1**	0	1	0	0
**n78-2**	0.0611	0	0.9389	0
**n78-3**	0	0	0	1
**n28**	0	0.1918	0.6126	0.1956

**Table 11 sensors-25-04978-t011:** Our proposed decision trees’ complexity.

Cell	Nodes	Depth
n258	14	3
n78-1	30	6
n78-2	150	15
n78-3	30	5
n28	730	12

**Table 12 sensors-25-04978-t012:** EE- and QoS-sensitive accuracy.

Cell	Node	Depth
n258	99.38%	98.28%
n78-1	94.81%	92.54%
n78-2	86.95%	98.52%
n78-3	94.53%	92.54%
n28	86.90%	87.54%

**Table 13 sensors-25-04978-t013:** Accuracy of our ML model in each cell.

	n258	n78-1	n78-2	n78-3	n28
**Training Accuracy**	97.33%	87.19%	77.33%	86.77%	76.67%
**Validation Accuracy**	97.66%	87.37%	76.53%	87.07%	75.03%

## Data Availability

Data is contained within the article or Appendix A.

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
