# Peer review of "A ML-Based Resource Allocation Scheme for Energy Optimization in 5G NR"

_sensors, 2025, doi:10.3390/s25164978_

Round 1
Reviewer 1 Report
Comments and Suggestions for Authors
In this work, author presents a machine learning based energy optimization framework for 5G New Radio systems. The authors propose a CART algorithm to dynamically allocate radio resources and implement a multi-domain sleep mode strategy, shutting down unnecessary resources across time, frequency, and antenna domains to optimize energy efficiency. A 5G testbed with an inter-band NR-DC layout is simulated, where a synthetic dataset of user traffic demands, based on a Radio Resource Control State Diagram is used to train and validate the ML model. The results indicate up to 42.3% reduction in energy consumption in normal traffic conditions, with satisfactory prediction accuracy (~84.7%) across different scenarios. Please check my following comments:
1) The study relies entirely on synthetic datasets. Although justified, it is recommended that the authors discuss more explicitly the limitations of synthetic data and outline a concrete plan for incorporating real-world traffic data in future work.
2) The authors should add some additional analysis showing how sensitive the energy savings and prediction accuracy are to changes in input parameters or user distributions.
3) The authors could add discussion on the interpretability of the CART-based model. How are key features influencing sleep state decisions? A feature importance analysis would enhance the manuscript.
4) It would be useful to include a computational complexity and scalability analysis for the proposed scheme, particularly regarding the feasibility of deploying the model in real-time 5G networks.
5) While five ML models were tested, no direct comparison to state-of-the-art resource allocation or heuristic methods (e.g., reinforcement learning-based schemes) is presented. Adding such comparisons would strengthen the manuscript.
6) Beyond accuracy, metrics such as F1-score, precision, recall, or confusion matrix could provide a more comprehensive assessment of the ML model's performance.
7) The manuscript mentions QoS thresholds but does not analyze QoS impact during resource reallocation. Including QoS performance metrics (latency, throughput) under sleep mode operations would be valuable.
Author Response
Comments 1: The study relies entirely on synthetic datasets. Although justified, it is recommended that the authors discuss more explicitly the limitations of synthetic data and outline a concrete plan for incorporating real-world traffic data in future work.
Response 1: Agreed. Due to the lack of real data, a brand-new traffic model based on a custom Radio Resource Control State Diagram (RRC-SD) has been developed from scratch, which also represents a contribution of our work. A wide and complete dataset was created by using that model to put our energy saving algorithm to a test. Following the recommendation of the reviewer, in section 7.3, we have added a discussion about the limitations of our synthetic data and how to overcome that point with a partnership with a mobile network operator to access real RAN logs or KPI information.
Comments 2: The authors should add some additional analysis showing how sensitive the energy savings and prediction accuracy are to changes in input parameters or user distributions.
Response 2: Thank you for pointing this out. We have taken into consideration that: while a full-scale sensitivity analysis could offer additional insight, we note that our UE demand scenarios already span a very wide range of spatial distributions, covering over 21,000 unique combinations. Given that our input granularity is in steps of 50 UEs per area, a reasonable sensitivity test would involve random shifts of ±25 UEs within each scenario. However, applying such fine-grained perturbations across all combinations would result in an extremely large and redundant test space. Conversely, evaluating only a few isolated scenarios would lack global significance and offer limited generalizability. Paying attention to the reviewer’s comments, we have accordingly added a thorough analysis of the proposed model in section 6.3, including the learning and validation process, the tuning parameters and so on.
Comments 3: The authors could add discussion on the interpretability of the CART-based model. How are key features influencing sleep state decisions? A feature importance analysis would enhance the manuscript.
Response 3: Agreed. We have, accordingly, added the interpretability and scalability analysis in section 6.3.1 and 6.3.2, where we have also analyzed the importance for each feature and the model complexity for each cell.
Comments 4: It would be useful to include a computational complexity and scalability analysis for the proposed scheme, particularly regarding the feasibility of deploying the model in real-time 5G networks.
Response 4: Agreed. We have, accordingly, added an insight about computational complexity and scalability analysis for the proposed scheme in section 6.3.2, including a table in page 12 with a comparison of computational efficiency for different ML models.
Comments 5: While five ML models were tested, no direct comparison to state-of-the-art resource allocation or heuristic methods (e.g., reinforcement learning-based schemes) is presented. Adding such comparisons would strengthen the manuscript.
Response 5: Agreed. We have considered the reviewer’s comments by digressing about the particularities of our algorithm in comparison to neural network-based resource allocation methods. This can be found in section 6.3.4.
Comments 6: Beyond accuracy, metrics such as F1-score, precision, recall, or confusion matrix could provide a more comprehensive assessment of the ML model's performance.
Response 6: Agreed. The traditional F1-score, precision, or recall may not be really applicable for our multi-class case. Hence, we have, accordingly, proposed two metrics: EE- and QoS-sensitive accuracy for our case based on the confusion matrix. A detailed discussion can be found in section 6.3.3.
Comments 7: The manuscript mentions QoS thresholds but does not analyze QoS impact during resource reallocation. Including QoS performance metrics (latency, throughput) under sleep mode operations would be valuable.
Response 7: Agreed. Direct measurements of QoS metrics such as latency depends on the architecture of the Radio Access Network, so it is beyond the scope of this paper. On the other hand, throughput can be indirectly obtained here from the total number of Data Radio Bearers assigned to each user. Precisely, in our paper, we have used a 99% DRB demand satisfaction threshold to select the most convenient sleep state, ensuring that the predicted state does not compromise UE service levels. Details can be found in page 11, in the last paragraph.

Reviewer 2 Report
Comments and Suggestions for Authors
The overall impression from the presented text is quite satisfactory.
The text itself is quite well-formatted. The authors’ logic is clear and somewhat easy to follow. But there are some issues with the submission that prevent me from recommending it for publication in the current form.
– The submission abounds with acronyms and abbreviations. Some are introduced to be used only once or twice. So, 1) please, add an acronym list at the beginning, for easiness of comprehension, and 2) critically revise the introduced acronyms, asking yourself a simple question: will you use it further, and do you really need to introduce it?
– Contribution of the submission is smeared all over the text. Moreover, authors’ contribution in light of the existing results is not evident. Please, solidify it and place it at the end of the introductory part (background part).
– It would be of great help to the reader to add a schematic representation of the proposed algorithm.
– The comparison of the proposed energy-saving scheme with the existing ones is absent. This is critical, since up to now several solutions exist. And it is not evident whether the proposed one has any advantages.
– As far as I understood, the authors created a synthetic database of UE traffic demands. This is an important result for the scientific community (in order to benchmark further developed methods with the one presented herein). So, if it is possible (due to funding issues), it would be good to make this database publicly available.
– While describing the machine learning stage, the authors were unacceptably too curt. Nothing is said about the learning and validation process (for instance, no learning and validation curves are present), the residual loss, tuning parameters, training time, etc.
– Moreover, it is not evident whether the trained model can perform in real time. So, some type of complexity analysis (and its comparison with the existing methods) is required.
Author Response
Comments 1: The submission abounds with acronyms and abbreviations. Some are introduced to be used only once or twice. So, 1) please, add an acronym list at the beginning, for easiness of comprehension, and 2) critically revise the introduced acronyms, asking yourself a simple question: will you use it further, and do you really need to introduce it?
Response 1: Agreed. We have, accordingly, added a list of abbreviations at the end of the paper, just before the references. Furthermore, we have revised the acronyms to make use of those strictly necessary.
Comments 2: Contribution of the submission is smeared all over the text. Moreover, authors’ contribution in light of the existing results is not evident. Please, solidify it and place it at the end of the introductory part (background part).
Response 2: Agreed. It is a good point so we have, accordingly, remarked the contributions of our work in a summary included in the Introduction, just before addressing the organization of the paper. It can be found in page 2.
Comments 3: It would be of great help to the reader to add a schematic representation of the proposed algorithm.
Response 3: Agreed. The ML algorithm here proposed uses the classification tree of a ML-CART model implemented in MATLAB, which involves a single-step, low-complexity classification task without multi-stage or iterative processes, hence the schematic presentation might be a little redundant. To help the reader to understand the algorithm, a more detailed ML model analysis has been included in section 6.3, with the interpreted visual decision tree for one particular case. The rest of visual decision trees have been included as supplementary materials to make the paper not be redundant or too extensive. We are confident that these may be enough to explain our model.
Comments 4: The comparison of the proposed energy-saving scheme with the existing ones is absent. This is critical, since up to now several solutions exist. And it is not evident whether the proposed one has any advantages.
Response 4: Agreed. We have, accordingly, added a discussion of comparison of existing method in section 6.3.4. Since our model is developed and evaluated under a custom-designed 5G RAN testbed, direct comparison with existing work can be rather difficult. However, to support further research and enable future comparisons, we have made our dataset publicly available, providing subsequent researchers the opportunity to benchmark their solutions against ours.
Comments 5: As far as I understood, the authors created a synthetic database of UE traffic demands. This is an important result for the scientific community (in order to benchmark further developed methods with the one presented herein). So, if it is possible (due to funding issues), it would be good to make this database publicly available.
Response 5: Agreed. That is a very good recommendation. We have, accordingly, published our dataset in Zenodo with the DOI: 10.5281/zenodo.16265621. Also, we have added a new section 6.3.5 to give more details about this open dataset. Furthermore, we added a discussion of limitation and future work of the synthetic dataset in section 7.3.
Comments 6: While describing the machine learning stage, the authors were unacceptably too curt. Nothing is said about the learning and validation process (for instance, no learning and validation curves are present), the residual loss, tuning parameters, training time, etc.
Response 6: Agreed. This is totally right. We have, accordingly, added a model analysis under section 6.3, including the learning and validation curves and the effect of some tuning parameters such as the maximum number of splits and the minimum number of leaf size. Regarding residual loss, since our model is a simple classification tree, direct residual analysis does not apply in the classical sense. Other than this, we have also proposed accuracy metrics suitable in our case for a better analysis and we have explained our model in section 6.3.3.
Comments 7: Moreover, it is not evident whether the trained model can perform in real time. So, some type of complexity analysis (and its comparison with the existing methods) is required.
Response 7: Agreed. We have, accordingly, added the complexity and scalability analysis in section 6.3.2 indicating how our model can perform in real life. We have also added the interpretability analysis in section 6.3.1, which visually presents our model for a better understanding. Moreover, a comparison with the existing methods is included in section 6.3.4.

Round 2
Reviewer 1 Report
Comments and Suggestions for Authors
The authors have revised the manuscript as per the comments..
Reviewer 2 Report
Comments and Suggestions for Authors
The authors have considered all the reviewer's comments and provided sufficient responses.